# *Lactobacillus rhamnosus* LRa05 Alleviates Constipation via Triaxial Modulation of Gut Motility, Microbiota Dynamics, and SCFA Metabolism

**DOI:** 10.3390/foods14132293

**Published:** 2025-06-28

**Authors:** Jingxin Zhang, Qian Li, Shanshan Liu, Ning Wang, Yu Song, Tao Wu, Min Zhang

**Affiliations:** 1State Key Laboratory of Food Nutrition and Safety, College of Food Science and Engineering, Tianjin University of Science and Technology, Tianjin 300457, China; 22843029@mail.tust.edu.cn (J.Z.); lss0105@mail.tust.edu.cn (S.L.); 15522987591@163.com (N.W.); 2Nutritious and Healthy Food Sino-Thailand Joint Research Center, Tianjin Agricultural University, Tianjin 300392, China; liqian1230@tjau.edu.cn (Q.L.); 13820951664@163.com (Y.S.)

**Keywords:** probiotic, constipation, intestinal mobility, gastrointestinal regulatory peptides, gut microbiota

## Abstract

Constipation, a widespread gastrointestinal disorder, imposes significant burdens on healthcare systems the and global health-related quality of life, yet current options remain suboptimal due to limited mechanistic understanding and efficacy limitations. Given the pivotal significance of the interactions between the gut microbiota and the host on governing bowel movement, we employed a multi-modal approach integrating animal experiments, ELISA, histopathology, qRT-PCR, GC-MS, and 16S rRNA metagenomics to evaluate the functional potential of *Lactobacillus rhamnosus* LRa05 against loperamide-induced constipation in mice. LRa05 treatment markedly alleviated constipation symptoms, as evidenced by reduced first black stool expulsion time, increased fecal moisture, and enhanced intestinal motility. Mechanistic investigations revealed that LRa05 balanced gastrointestinal regulatory peptides. It also downregulated aquaporin (AQP4/AQP8) mRNA levels and activated the SCF/C-Kit signaling pathway. These effects contributed to the restoration of intestinal peristalsis. Furthermore, LRa05 rebalanced gut microbiota composition by enriching beneficial, including *Alloprevotella* and *Lachnospiraceae NK4A136*, key SCFA producers. Thus, LRa05 could boost short chain fatty acid (SCFA) production, which is vital for stimulating intestinal motility, improving mucosal function, and relieving constipation. These findings demonstrated that LRa05 could mitigate constipation through a multi-target mechanism: regulating motility-related gene transcription, restructuring the microbial community, balancing gastrointestinal peptides, repairing the colonic mucosa, and promoting SCFAs for fecal hydration. Our study positions LRa05 as a promising probiotic candidate for constipation management.

## 1. Introduction

Constipation, a prevalent global digestive disorder, is characterized by difficult defecation, dry stools, and infrequent bowel movements, significantly diminishing patients’ quality of life [1]. Untreated constipation can progress to severe complications, including various gastrointestinal disorders, and an elevated risk of colorectal cancer. Chronic cases without effective intervention, can cause irreversible intestinal damage, escalating health risks and healthcare burdens [2]. This underscores the urgent need for therapeutic strategies that target the underlying mechanisms of constipation to mitigate its cascading pathological effects.

Pathophysiologically, constipation involves intestinal microbiota imbalance, abnormal gastrointestinal hormones secretion, gut nervous system (GNS) dysfunction, and impaired intestinal motility. Dysbiosis reduces metabolite production, affecting motility and water absorption [3]. Hormonal abnormalities (such as motilin and substance P) weaken the intestinal contractility [4]. The gut nervous system disorders impede intestinal movement regulation [5]. The SCF/C-kit pathway imbalance impairs interstitial cells of Cajal (ICC), the intestinal “pacemakers”, slowing transit [6]. Colonic aquaporins (AQP 3/4/8/9) dysregulate water transport, hardening stools [7]. Upstream triggers (e.g., genetics, epigenetics) and pathway interactions remain unclear, including microbiota-derived metabolite modulation of AQP expression.

Currently, the treatment of constipation predominantly relies on laxatives and gastrointestinal motility drugs. Laxatives, which typically contain chemicals that enhance the frequency of bowel peristalsis, are the most commonly prescribed medications for constipation [8]. They facilitate defecation by boosting the delivery of water and electrolytes to the intestinal mucosa and softening hardened stools [9]. These drugs enhance bowel peristalsis but only offer temporary relief, failing to restore short-chain fatty acids (SCFAs) or intestinal barrier function. Despite a $4.2 billion market, conventional therapies show a 67% recurrence rate and 32% adverse event rate (NIH data), driving demand for multi-targeted approaches.

Probiotics have emerged as promising functional agents for gastrointestinal disorders, including constipation. The constipation-specific probiotic market grows at 11.2% annual compound growth rate (CAGR), with strain-specific products commanding a 35–40% price premium. Probiotics work through multiple mechanisms, such as modulating gut microbiota, promoting intestinal movement, repairing the mucosal barrier, and regulating the gut nervous system [10]. *Lactobacillus rhamnosus* has gained prominence as a potent modulator of the gut microbiota, due to its excellent adhesive properties, immunomodulatory capabilities, and ability to maintain intestinal homeostasis [11,12]. This probiotic has demonstrated significant efficacy in restructuring the gut microbial community by promoting the development of beneficial bacteria while curbing the expansion of harmful microbes [13]. It also exhibits anti-inflammatory characteristics, increases SCFA levels, and modulates the expression of tight-junction proteins (ZO-1, occludin, claudin), thereby strengthening the intestinal barrier and optimizing epithelial permeability [14]. Given the vital role of the intestinal barrier in constipation pathogenesis, these findings suggest that *Lactobacillus rhamnosus* LRa05 holds great potential for the treatment of this condition. By integrating mechanistic understanding with commercial viability, LRa05 represents a new frontier in precision nutrition probiotics, marking a shift from merely managing symptoms to proactively intervening in the pathophysiological processes of constipation [15].

To evaluate the efficacy of *Lactobacillus rhamnosus* LRa05, a loperamide-induced murine model of constipation was utilized. The study comprehensively analyzed changes in intestinal motility patterns, characteristics of fecal output, and serum levels of regulatory gastrointestinal peptides. Additionally, the impact of the strain on SCFA production, the SCF/C-kit signaling pathway, and the expression of aquaporins (AQP4 and AQP8) was investigated to gain a mechanistic understanding at the molecular level in colonic tissue. Complementary 16S rRNA gene sequencing was performed to identify constipation-associated dysbiosis in the gut microbiota. The findings established a mechanistic framework for LRa05’s multi-target modulation of constipation, serving as a scientific foundation for its application in functional food development.

## 2. Materials and Methods

### 2.1. Chemicals

*Lactobacillus rhamnosus* LRa05 probiotic strain was supplied by Wecare Probiotics Co., Ltd. (Suzhou, China). Loperamide hydrochloride capsules (2 mg/capsule) were obtained from Xian Janssen Pharmaceutical Ltd. (Xi’an, China). Mosapride citrate tablets (5 mg/tablet) were purchased from Yabao Pharmaceutical Group Co., Ltd. (Shanghai, China). All other chemical reagents were of analytical grade (≥99% purity) unless otherwise specified.

### 2.2. Strain Activation and Bacterial Suspension Preparation

*Lactobacillus rhamnosus* LRa05 was activated and propagated in MRS broth at 37 °C for 16 h. Serial subculturing was performed by inoculating 2% *v/v* into fresh MRS broth, with a minimum of three passages to ensure stable growth characteristics. Bacterial cells were collected by centrifugation (6000 rpm, 12 min), washed twice with sterile saline (0.9% NaCl), and resuspended to final concentrations of 6 × 10^8^ CFU/mL (low dose) and 1.2 × 10^9^ CFU/mL (high dose). The bacterial suspension was prepared daily and used immediately for gavage administration during animal experiments.

### 2.3. Laboratory Animals and Intervention Procedures

Male SPF-grade Balb/c mice (*n* = 50, 20 ± 2.0 g) were sourced from SPF Biotechnology Co., Ltd. (Beijing, China, Certification SCXK [Jing] 2019-0010) and maintained in specific pathogen-free conditions at TUST Animal Research Center (23 ± 1 °C, 12 h/12 h light cycle, ad libitum access to sterilized feed/water). The sample size (*n* = 10 per group) was determined based on prior studies demonstrating that this cohort size could detect biologically significant differences in gastrointestinal motility parameters [16,17]. Through stratified randomization after 7-day acclimatization, cohorts were established as the native control group (NC), the model group (MC), the positive control group (PC), the low-dose LRa05 (LRa05.L) group, and the high-dose LRa05 (LRa05.H) group. Stratification was performed based on baseline body weight and fecal pellet output to ensure comparable distributions across groups. Randomization sequence generation was computerized using R software (version 4.2.1), and animals were allocated to groups by an investigator not involved in outcome assessment. During the second week, constipation models were induced in all groups except NC by gavage administration of loperamide hydrochloride (10 mg/kg body weight). Starting from the third week, the MC group received daily 0.9% sterile saline (0.2 mL) via gavage, while the PC group was treated with mosapride citrate suspension at 5 mg/kg using the same administration route. For the probiotic intervention groups, LRa05.L (6 × 10^8^ CFU/mL) and LRa05.H (1.2 × 10^9^ CFU/mL) concentrations were calculated based on human probiotic intake guidelines (1–10 × 10^9^ CFU/day). Allometric scaling principles (body surface area normalization) were applied to convert the human-equivalent doses to murine equivalents. The final formulations were adjusted to deliver 6 × 10^8^ CFU (LRa05.L) and 1.2 × 10^9^ CFU (LRa05.H) per 0.2 mL gavage volume for mice weighing 20–25 g, which is consistent with established rodent probiotic dosing standards [18]. Outcome assessors were blinded to group assignments throughout the experiment. Treatments were coded and administered by a separate investigator, and data analysis was performed using de-identified datasets. Blinding was maintained until statistical analysis was completed.

The animal study protocol was approved by the Institutional Animal Care and Utilization Committee of TUST (protocol code 2,023,010 and date of approval 15 March 2023). All procedures adhered to the ARRIVE guidelines (Animal Research: Reporting of In Vivo Experiments) and the National Institutes of Health Guide for the Care and Use of Laboratory Animals. Animals were euthanized via CO_2_ inhalation followed by cervical dislocation under deep anesthesia, and all efforts were made to minimize suffering.

### 2.4. Sample Collection

On day 22 (including the adaptation period), all mice were humanely euthanized. Prior to euthanasia, fresh fecal pellets were cryopreserved within 2 min of spontaneous defecation using liquid nitrogen immersion prior to −80 °C archival (Thermo Scientific™, Waltham, MA, USA, Forma™ 900 series). Mice underwent a 12-h fasting period (with ad libitum water access) prior to terminal procedures. Following retro-orbital venous plexus blood collection into lithium heparin-coated vacutainers, specimens underwent coagulation at 25 °C ± 0.5 °C for 60 min. Serum isolation was achieved through dual-phase centrifugation (Primary: 4000× *g*, 15 min, 4 °C; Secondary: 12,000× *g*, 2 min, 4 °C) using a refrigerated centrifuge (Model TDL-5-A, Xiangyi Centrifuge, Changsha, China). After blood collection, mice were euthanized by cervical dislocation under isoflurane anesthesia. Colon tissues were rapidly excised, rinsed in ice-cold PBS, and divided for histopathological evaluation (fixed in 10% neutral-buffered formalin) and RNA extraction (snap-frozen in liquid nitrogen for qPCR analysis).

### 2.5. Body Mass, Feeding Behavior, Fecal Transit, and Intestinal Motility Assays

Body mass dynamics and ad libitum food consumption were continuously monitored using a computer-assisted monitoring system (PhenoMaster NG, TSE Systems China, Beijing, China).

On Day 21 (including the adaptation period), mice were fasted for 12 h (water ad libitum) before gavage administration of 0.2 mL activated charcoal solution (5% *w/v* activated charcoal + 10% *w/v* gum arabic). Time to first black stool (min) was recorded using a digital timer. Fresh feces were collected, weighed (wet weight, W1), and dried at 105 °C to constant weight (dry weight, W2) [19]. Fecal water content (%) was expressed as the following an Equation:Fecal water content (%) = [(W1 − W2)/W1] × 100(1)

On Day 22 (including the adaptation period), intestinal propulsion was evaluated using a modified charcoal transit assay [20]. After 12 h fasting, mice received 0.2 mL charcoal solution. At 25 min post-gavage, animals were euthanized, and intestines excised. We measured the total small intestine length (L1, cm) and charcoal front distance (L2, cm). Propulsion rate was expressed as the following Equation:Propulsion rate (%) = L2/L1 × 100(2)

### 2.6. Quantitative Profiling of Gastrointestinal Regulatory Peptides

Circulating levels of key neuroendocrine regulators, gastrin (GAS), motilin (MTL), substance P (SP), somatostatin (SS), vasoactive intestinal peptide (VIP), endothelin-1 (ET-1), and 5-hydroxytryptamine (5-HT), were quantified using high-sensitivity ELISA kits (Sigma-Aldrich (Shanghai)Trading Co., Ltd., Shangai, China). Optical density (OD) measurements were conducted at 450 nm using a SpectraMax i3x microplate reader (Molecular Devices, San Hose, CA, USA), with a reference wavelength of 630 nm to correct for non-specific absorbance. All assays strictly adhered to the manufacturer’s validated protocols, including pre-incubation of samples at 4 °C overnight to optimize antigen-antibody binding. Inter-assay coefficients of variation (CVs) were maintained below 8% across all analytes, demonstrating assay reproducibility.

For quantitative analysis of enteroendocrine mediators, a standardized ELISA protocol was employed. Briefly, 50 μL serum aliquots were diluted 1:5 in assay buffer (PBS, pH 7.4) containing 0.05% Tween-20 and 1% bovine serum albumin (BSA) to minimize matrix effects. Samples were loaded in triplicate onto pre-coated 96-well plates and incubated at room temperature for 2 h. Parallel standard curves were generated using seven-point serial dilutions of calibrators (ranging from 0.1 to 100 ng/mL), with blank controls (assay buffer alone) run in quadruplicate for background subtraction. After washing with PBS-Tween (0.05%) buffer, bound analytes were detected using HRP-conjugated secondary antibodies and developed with tetramethylbenzidine (TMB) substrate. Reactions were terminated with 2 M sulfuric acid, and OD values were immediately recorded. Data were analyzed using SoftMax Pro 7.0 software, with concentrations calculated from four-parameter logistic regression curves.

### 2.7. Histopathological Evaluation of Colonic Mucosa

To evaluate LRa50 probiotic’s impact on intestinal barrier integrity in murine constipation models, a systematic histomorphometric analysis of distal colonic architecture was conducted. Post-euthanasia, 2 cm colonic segments underwent immediate fixation in 4% paraformaldehyde (Servicebio, Wuhan, China) for >12 h to preserve cytoarchitecture. Specimens underwent standard paraffin embedding, followed by sectioning at 4 μm (CV <5%) using a rotary microtome (HistoCore MULTICUT, Leica Microsystems). Section flattening was achieved through controlled thermal equilibration (40–42 °C water bath, 120 s) followed by adhesive mounting on pre-treated slides.

Deparaffinization was achieved through two 10-min xylene immersions, followed by gradient ethanol rehydration (100% to 70%). Hematoxylin-eosin (H&E) staining proceeded as follows: 10-min hematoxylin (Solaibao Reagent Co., Ltd., Beijing, China) nuclear staining, 5-s differentiation with 1% HCl-ethanol, and blueing via running water. Eosin (Solaibao Reagent Co., Ltd., Beijing, China) counterstaining (3 min) highlighted cytoplasmic structures. Specimens were dehydrated in graded alcohols, cleared in xylene (2 × 8 min), and coverslipped with a cover glass.

Slides were observed under an upright microscope (BX53, Olympus, Tokyo, Japan) at 200× magnification. Quantitative comparisons of the mucus layer thickness between treatment and control groups were documented for statistical evaluation.

### 2.8. Quantitative Analysis of Fecal Short-Chain Fatty Acids (SCFAs)

Quantitative analysis of SCFAs was determined through optimized gas chromatography-mass spectrometry (GC-MS) methodology modified from established protocols [21]. Freeze-dried fecal specimens (30.0 ± 0.5 mg) underwent acidified extraction using 800 μL ultrapure water, 200 μL sulfuric acid (50% *v/v*, HPLC grade), and 1 mL diethyl ether (GC-MS grade). The homogenate was vortex-mixed (2500 rpm, 3 min) followed by phase separation in an ice bath (4 °C, 30 min). After centrifugation (10,000× *g*, 10 min, 4 °C), the organic layer was dehydrated with anhydrous sodium sulfate (0.25 g) and clarified through secondary centrifugation (10,000× *g*, 15 min). Processed extracts were transferred to amberized GC vials with PTFE-lined septa for immediate analysis on a GC-MS system (Thermo Scientific ISQ 7000 GC/MS, USA).

Separation was performed using a Thermo Scientific™ TG-WAXMS polar capillary column (30 m × 0.25 mm i.d., 0.25 μm film thickness). The initial oven temperature program was set at 80 °C for 2 min, followed by a temperature gradient increase at 10 °C/min to 240 °C with a 5-min hold. In electron ionization (EI) mode, the ionization energy was fixed at 70 eV, and full-scan mass spectrometry was conducted over the *m/z* range of 40–450. For short-chain fatty acid analysis, characteristic fragment ions *m/z* 43 (Acetate), *m/z* 74 (Propionate), *m/z* 60 (Butyrate), *m/z* 87 (Isobuturic), *m/z* 101 (Valeric and Isovaleric), were selected as quantitative ions to ensure specific detection and accurate quantification of target compounds.

### 2.9. Gene Expression Analyzed via Quantitative Reverse Transcription PCR (qRT-PCR)

To characterize mRNA expression levels of colonic genes, including aquaporin 4 (AQP4), aquaporin 8 (AQP8), stem cell factor (SCF), and C-Kit, qRT-PCR analysis was conducted using validated methodologies [22]. Total RNA was extracted from snap-frozen intestinal specimens using TRIzol™ Reagent (Invitrogen, Carlsbad, CA, USA) with mechanical homogenization (Bertin Precellys^®^ 24, Thiron-Gardais, France). RNA integrity was verified through microfluidic electrophoresis (Agilent 2100 Bioanalyzer, Santa Clara, CA, USA, RIN > 8.0). First-strand cDNA synthesis was performed using the PrimeScript™ RT Master Mix (Takara, Kusatsu, Japan) with genomic DNA elimination, following the manufacturer’s protocol to synthesize cDNA.

The qPCR amplification was conducted utilizing TB Green^®^ Premix Ex Taq™ II (Takara, Shiga, Japan) on a CFX Connect™ Real-Time System (Bio-Rad, Hercules, CA, USA) with the following thermal profile: initial denaturation at 95 °C for 30 s, followed by 40 cycles of 95 °C for 5 s and 60 °C for 10 s.

Relative gene expression was calculated using the 2^−ΔΔCT^ method with β-actin as the endogenous control. Each sample was analyzed in triplicate, and average Ct values were obtained. ΔCt was determined by subtracting the β-actin Ct value from the target gene Ct value (ΔCt = Ct(target) − Ct(β-actin)). ΔΔCt was calculated by subtracting the ΔCt value of the blank control group from that of the treatment group, following this formula:ΔΔCt = ΔCt(treatment) − ΔCt(control)(3)

Final relative expression levels were expressed as 2^−ΔΔCt^ values. Gene-specific primers used in this study are listed in Table 1.

### 2.10. Gut Microbiota Analysis

Mouse fecal samples (200 mg) were aseptically collected and immediately stored at −80 °C to preserve microbial DNA integrity. Samples were homogenized using sterile mortar and pestle to ensure uniform microbial cell distribution and improve DNA extraction efficiency. Genomic DNA was extracted using the UltraClean Fecal DNA Isolation Kit (MoBio Laboratories, Carlsbad, CA, USA), involving mechanical lysis and column-based purification.

The V3-V4 hypervariable regions were amplified and sequenced on the Illumina NovaSeq (San Diego, CA, USA) 6000 platform (2 × 250 bp PE, >100,000 reads/sample). The raw paired-end reads were processed through a comprehensive bioinformatics pipeline. Initially, primer sequences were removed using Cutadapt v2.10. Subsequently, the reads underwent quality filtering with a quality score threshold of Q ≥ 30 and were truncated at 250 base pairs. The processed reads were then subjected to denoising and chimera removal using DADA2 v1.24, followed by the generation of amplicon sequence variants (ASVs).

Taxonomic annotation was performed against the Greengenes database (Silva version 138.1) using QIIME2 v2020.8 (Illumina, Inc., USA). Phylogenetic relationships were inferred via MAFFT v7.490 multiple sequence alignment. Relative abundances at taxonomic levels (phylum to species) were normalized using cumulative sum scaling.

Alpha diversity indices (Shannon, Chao1, Simpson, observed OTUs) were calculated to assess within-sample diversity. Beta diversity was evaluated using Bray–Curtis dissimilarity and visualized via non-metric multi-dimensional scaling (NMDS). Linear discriminant analysis effect size (LEfSe) was applied to identify differentially abundant taxa between groups (LDA score > 3.0). Spearman’s rank correlation coefficients were calculated to assess associations between dominant phyla and physiological parameters, with significance adjusted via Benjamini–Hochberg correction. Based on 16S rRNA sequencing data and reference genome datasets, PICRUSt2 (Phylogenetic Investigation of Communities by Reconstruction of Unobserved States) was utilized to predict functional pathways (KEGG orthologs and metabolic modules). This was achieved by reconstructing the gene content of unobserved microbial taxa, enabling inference of the metagenomic functional potential of the community. Predicted functional profiles were normalized to account for sampling depth variations and annotated to assign biological interpretations to the inferred functions.

To confirm the homogeneity of the groups at the microbial level. One animal was randomly selected from each group for intestinal flora sequencing analysis before the intervention. The results showed no significant differences in key microbial taxa (*p* > 0.05).

### 2.11. Statistical Analysis

Data are presented as mean ± standard error of the mean (SEM). Normality and homogeneity of variances were confirmed prior to analysis. Normality of data distribution was assessed using the Shapiro-Wilk test (α = 0.05) for each experimental group, and homogeneity of variances was confirmed via Bartlett’s test prior to parametric analysis. For datasets meeting normality and variance assumptions (*p* > 0.05 for both tests), one-way analysis of variance (ANOVA) followed by Duncan’s multiple range test (with Bonferroni correction for multiple comparisons) was applied to evaluate significant differences among the ten experimental groups. For datasets failing normality or variance tests, Kruskal-Wallis non-parametric ANOVA followed by Dunn’s post hoc test with FDR correction was used instead. All statistical analyses and graph generation were performed using GraphPad Prism 10.1 (GraphPad Software, San Diego, CA, USA), with statistical significance (*p* < 0.05, two-tailed tests). Effect sizes (*η*^2^ for ANOVA, r for non-parametric tests) were calculated to quantify the magnitude of differences between groups.

## 3. Results and Discussion

### 3.1. LRa05-Mediated Alterations in Excretory Patterns and Intestinal Functionality

Constipation pathophysiology manifests as colonic dysmotility, diminished defecation frequency, and prolonged intestinal transit latency. Compared to the NC group, the constipation model (MC group) demonstrated marked stool dehydration, exhibiting a 38.2% reduction in fecal water content versus native controls (NC, 61.5 ± 1.7%; MC, 48.2 ± 2.1% [g H_2_O/g stool], *p* < 0.001; Figure 1a). The duration until the initial occurrence of black feces was notably prolonged (NC, 91.8 ± 7.1; MC, 157.8 ± 5.1 min, *p* < 0.001; Figure 1b). This indicated that the mice clearly experienced defecation difficulties after the establishment of the constipation model.

After the oral administration of *Lactobacillus rhamnosus* LRa05, mouse feces had significantly higher moisture content than the MC group (*p* < 0.01, Figure 1a). The low and high-dose groups (LRa05.L and LRa05.H) reached 52.17% and 54.83%, respectively. Moreover, the time to first black feces was significantly shorter (*p* < 0.001 vs. MC group). The LRa05.L group showed a 22.9% decrease (121.7 ± 8.0 min), and LRa05.H showed a 30.1% decrease (110.3 ± 5.6 min). These results implied that *Lactobacillus rhamnosus* LRa05 has the potential to optimize defecation-related metrics in mice suffering from constipation. Specifically, it could stimulate the defecation process and boosts the moisture content of feces. In prior investigations, it has been convincingly shown that Bifidobacterium strains are capable of substantially accelerating intestinal peristalsis, preserving the water content within feces, and reducing the time taken for defecation. These results suggested LRa05 could improve constipation by increasing fecal hydration and accelerating defecation, aligning with clinical goals of softening stools and shortening defecation intervals [19].

Disrupted intestinal motility may extend fecal retention in the colon, resulting in excessive water reabsorption from intestinal contents and the subsequent formation of dry, hardened stools. As depicted in Figure 1c and Appendix A, the carbon powder transport assay demonstrated that MC group mice had significantly lower intestinal propulsion. Their propulsion rate was 39.3 ± 3.4%, much lower than that of the NC group (64.2 ± 3.0%, *p* < 0.001). Notably, LRa05 improved intestinal propulsion in a dose-dependent manner. LRa05.L had a rate of 44.8 ± 2.8%, and LRa05.H reached 48.7 ± 2.3%. The high dose showed a particularly strong effect, differing significantly from the MC group (*p* < 0.001).

These findings support LRa05’s potential application for improving constipation via enhanced gastrointestinal transit efficiency, functionally analogous to clinical prokinetic drugs like mosapride, but through milder biological pathways [23].

Appendix A delineated longitudinal alterations in body mass and food intake profiles across experimental cohorts. Univariate analysis of variance (ANOVA) revealed the trend of weight loss in intervention groups relative to baseline controls (NC group) during the modeling phase (*Δ*mass = 0.62 − 0.86 vs. 1.02 g). The administration of *Lactobacillus rhamnosus* LRa05 (LRa05.H, 2.4 × 10^9^ CFU/d) restored adipose tissue homeostasis, leading to mass accretion indices comparable to those of the NC group (*Δ*mass = 0.58 vs. 0.81 g, *p* > 0.05). By contrast, the model group (MC group) exhibited negligible somatic recovery (*Δ*mass = 0.08 g).

A quantitative investigation disclosed that during the course of treatment, the food intake of the MC group’s constipated mice was diminished by 25% in comparison with the NC group. Subsequent probiotic intervention with *Lactobacillus rhamnosus* LRa05 restored circadian feeding rhythms, achieving a 99% recovery of food intake. The findings indicated that LRa05 could alleviate the weight loss and food intake reduction caused by constipation in mice to a certain extent. This might be related to the dual-action mechanism of LRa05. For example, it enhanced ghrelin-mediated orexigenic signaling while suppressing leptin-induced appetite inhibition. This helped improve constipation-induced reductions in food intake and body weight in mice [24].

### 3.2. Impact of LRa05 on the Serum Concentrations of Gastrointestinal Regulatory Peptides

To investigate whether *Lactobacillus rhamnosus* LRa05 is capable of modulating the levels of gastrointestinal hormone peptides, serum levels of MTL, GAS, SP, SS, ET-1, VIP, and 5-HT were quantified. In this context, MTL, GAS, and SP, as excitatory neurotransmitters, enhanced gastrointestinal motility, while ET-1, SS and VIP, as inhibitory ones, decreased it. In the intestine, a dynamic equilibrium exists between inhibitory peptides and excitatory peptides, which respectively regulate intestinal secretion and absorption functions. 5-HT bidirectionally regulates gastrointestinal motility. Deficiency causes constipation, while excess leads to diarrhea.

The results, as depicted in Table 2, revealed a significant reduction in the concentrations of excitatory neurotransmitters, namely SP, MTL, and GAS, when comparing the MC group with the native NC group (*p* < 0.001). Specifically, the concentration of MTL, GAS, and SP measured 395.0 ± 8.6 pg/mL, 66.4 ± 2.2 pg/mL, and 304.1 ± 8.0 pg/mL, respectively. In stark contrast, in the MC group, these levels decreased to 264.8 ± 13.5 pg/mL, 36.8 ± 2.4 pg/mL, and 176.4 ± 8.5 pg/mL, respectively. This neurotransmitter depletion reflects the gut nerve-ICC injury-inflammation vicious cycle seen in clinical constipation, where ICC damage exacerbates colonic inertia [25]. Upon the intervention of LRa05, when compared with the concentration in the MC group, the high-dose LRa05 could significantly restore the concentration of GAS and SP (*p* < 0.001). Specifically, the level of GAS increased by 31.1% (from the baseline value in the MC group to 48.2 ± 1.5 pg/mL), and the level of SP increased by 26.7% (from the baseline value in the MC group to 223.5 ± 5.6 pg/mL). This suggested that *Lactobacillus rhamnosus* LRa05 may enhance gastrointestinal motility by restoring excitatory neurotransmission—a novel probiotic approach to clinical neurotransmitter-targeted therapy [26].

Compared with the NC group, the levels of inhibitory neurotransmitters VIP, SS, and ET-1 (Table 2) in the serum of the MC group were significantly elevated (*p* < 0.001), reaching 174.1 ± 5.7 pg/mL, 135.8 ± 5.5 pg/mL, and 216.7 ± 10.0 pg/mL, respectively. This dominance of inhibitory peptides likely promotes excessive intestinal water absorption, contributing to dehydration-associated constipation in clinical settings. In constipated mice, higher levels of inhibitory neurotransmitters likely reflect a dysfunctional interaction. This involves overactive neuroimmune responses, compensatory dehydration mechanisms, and disrupted communication between GNS and ICC cells. Targeting VIP/SS/ET-1 signaling or restoring neuro-glial-ICC balance could break this cycle and alleviate dehydration-related constipation. After the intervention of LRa05, the levels of VIP, SS, and ET-1 in the high-dose group were significantly reduced by 18.06%, 36.44%, and 38.30% (from the baseline value in the MC group to 142.6 ± 7.1 pg/mL, 86.3 ± 2.5 pg/mL, 133.7 ± 7.9 pg/mL, *p* < 0.01), respectively. These findings suggest LRa05 rebalances gut neurotransmission by suppressing inhibitory pathways. This likely happens by blocking enzymes or transcription factors that make these inhibitors, or by adjusting secretory cell activity. This intervention potentially synergizes with clinical receptor antagonist strategies but via safer network regulation [25]. Through the dual modulation of peptide networks, LRa05 could concurrently promote intestinal fluid secretion, alleviate luminal dehydration, and soften fecal consistency. This multi-targeted intervention tackles the root cause of constipation—mucosal water-salt balance disruption—by restoring epithelial transport and neuromodulatory signals.

5-HT levels were severely reduced in constipated mice (NC, 5.68 ± 0.20 pg/mL; MC, 2.24 ± 0.06 pg/mL; *p* < 0.001; Table 2). The collapse of 5-HT levels in constipated mice reflects enterochromaffin cell failure and microbiota-GNS-ICC axis disruption in clinical constipation. Restoring 5-HT signaling (e.g., probiotics, 5-HT4 agonists, or tryptophan supplementation) could break the cycle of motility arrest and dehydration. Similarly, moxapride applied in the PC group partially restored 5-HT (PC, 5.07 ± 0.16 pg/mL; LRa05.L, 2.90 ± 0.08 pg/mL; LRa05.H, 3.66 ± 0.11 pg/mL; *p* < 0.01 or *p* < 0.001; Table 2). Though less effective than the drug moxapride, LRa05 likely boosts 5-HT signaling via tryptophan hydroxylase (TPH1) activation or microbial metabolite modulation, supporting 5-HT4 agonist-probiotic combination therapy in clinical practice [27]. Specifically, 5-HT binds to 5-HT4 receptors on gut neurons, stimulating acetylcholine release and enhancing smooth muscle contraction, thereby accelerating intestinal transit. Conversely, chronic 5-HT elevation (e.g., in diarrhea) desensitizes 5-HT receptors (e.g., 5-HT4), paradoxically reducing motility. Furthermore, excess 5-HT recruits mucosal immune cells (e.g., mast cells) to release inflammatory mediators (e.g., histamine), indirectly inhibiting smooth muscle activity.

Collectively, *Lactobacillus rhamnosus* LRa05 exerts its constipation-relieving effects through the multi-faceted modulation of gut hormone peptides, with a primary focus on the GBA pathway. These effects are hypothesized to be mediated via direct interactions with intestinal epithelial cells, microbial metabolites, or host signaling cascades, underscoring its potential as a functional probiotic. It demonstrated that *Lactobacillus rhamnosus* strains could selectively elevate MTL and 5-HT levels via distinct molecular routes, thereby improving gastrointestinal transit in constipated murine models [28]. Gut microbes can directly or indirectly make and regulate brain chemicals, like 5-HT and SP, affecting the function of both the gut nervous system and central nervous system (CNS). Mechanistically, LRa05 boosts 5-HT production by rewiring tryptophan metabolism, and promotes SP release, triggering vagal nerve signals to brainstem areas like the nucleus tractus solitarius. Furthermore, through the suppression of pro-inflammatory mediators ET-1 and VIP, LRa05 restored microbiota–metabolite–host signaling integrity. Specifically, ET-1 reduction mitigated intestinal ischemia and inflammatory responses, whereas VIP downregulation diminishes its inhibitory effects on GNS excitability, rebalancing smooth muscle contractility. LRa05’s ability to balance multiple neurotransmitters makes it a promising treatment, potentially outperforming single-target drugs like 5-HT agonists.

Notably, the positive control group (treated with mosapride citrate) demonstrated more pronounced alterations in gastrointestinal peptide levels (Table 1) but did not outperform the LRa05 group in constipation relief (Figure 1a), underscoring the multi-factorial nature of constipation regulation. This paradox underscores that gastrointestinal peptides alone do not fully govern constipation pathophysiology, and additional mechanisms, particularly those modulated by probiotics, are critical for symptom resolution. It is important to note that while these inferences are supported by serum ELISA data (e.g., altered 5-HT, SP, and MTL levels), they remain speculative without protein-level validation (e.g., Western blot or tissue immunostaining) and cellular source identification (enteroendocrine vs. neuronal origin).

### 3.3. Mediation of Colonic Histopathological Integrity by LRa05

To evaluate the restorative potential of *Lactobacillus rhamnosus* LRa05 on colonic integrity, systematic histopathological analyses in loperamide-induced constipated mice were performed. Hematoxylin-eosin (HE) stained colon sections were examined at 200× magnification to characterize tissue architecture (Figure 1b) and quantify indices (Figure 1c).

The NC group displayed intact colonic morphology featuring orderly intestinal villi alignment and abundant goblet cell populations within crypts. In contrast, the MC group exhibited hallmark colonic pathological alterations. As shown in Figure 1b, these included disrupted epithelial polarity, mucosal atrophy with muscularis dissociation (green arrow), focal inflammatory infiltrates (blue arrow), and significant muscularis propria thinning (red arrow). Quantitative histomorphometric analysis demonstrated severe constipation-induced colonic structural degeneration (Figure 1c). The smooth muscle layer thickness plummeted by 67.4% in the model group (MC, 45.2 ± 0.9 μm) compared to native controls (NC, 138.6 ± 6.7 μm; *p* < 0.001), indicating compromised contractile capacity. The crypt depth and villus height reduced by 65.1% and 45.6% (NC, 93.6 ± 0.8 and 181.3 ± 2.9 μm vs. MC, 32.7 ± 1.3 and 98.6 ± 2.1 μm). These metrics collectively reflect profound mucosal–muscular dissociation and impaired barrier functionality, hallmarks of chronic constipation pathophysiology. These pathologies mimic clinical constipation features like mucosal atrophy and inflammatory cell infiltration observed in colonoscopy [29].

These pathological changes in the MC group likely arose from constipation-related physiological impairments. Slow fecal transit exerts abnormal mechanical stress on the colonic mucosa, which disrupts epithelial architecture and induces muscularis propria dissociation. Prolonged contractile hypoactivity in constipation progressively thins the muscular layer. Concurrently, compromised epithelial barrier function triggers focal inflammatory infiltrates via immune cell migration. The disorganization of goblet cells—critical for mucus production that maintains intestinal lubrication—further exacerbates mucosal dysfunction. Impaired mucus secretion culminates in dry, hardened stools, perpetuating a vicious cycle of colonic tissue damage [30].

Notably, LRa05 administration dose-dependently reversed pathological remodeling. The low-dose group (LRa05.L) restored colonic architecture to levels comparable with the positive control (PC) group, demonstrating an intermediate recovery of mucosal folds (PC, 76.3 ± 12.0 μm and 72.2 ± 6.8 μm) and goblet cell density. The high-dose group (LRa05.H) achieved the re-established goblet cell stratification and attenuated inflammatory infiltration (Figure 1b), which was critical for intestinal lubrication and barrier defense. Colon wall thickness surged 1.8-fold versus the model group (MC, 45.2 ± 0.9 μm vs. LRa05.H, 81.2 ± 0.9 μm; Figure 1c), reinstating contractile vigor. The crypt depth and villus height achieved 2.7-fold recovery, and 64.1% restitution (MC, 32.7 ± 1.3 and 98.6 ± 2.1 μm vs. LRa05.H, 87.8 ± 2.0 and 161.7 ± 1.1 μm; Figure 1c), enhancing absorptive surface area. These findings demonstrated LRa05’s capacity to improve intestinal barrier dysfunction in clinical constipation, complementing mucosal protectant therapies. This functional profile parallelled reports that *Lactobacillus rhamnosus* LRJ-1 alleviated distal colonic injury via goblet cell regeneration and muscularis reinforcement (*p* < 0.05).

The observed effects of *Lactobacillus rhamnosus* LRa05 were likely mediated by dual mechanisms. These involved augmenting gastrointestinal motility and the biosynthesis of bioactive metabolites. The two mechanisms acted synergistically to mitigate constipation-induced colonic injury. Reduced fecal retention alleviates mechanical stress on the mucosa and limits exposure to cytotoxic metabolites (e.g., ammonia, secondary bile acids) and pathogenic bacteria-derived endotoxins. This mitigates epithelial barrier disruption and inflammatory cascades, preserving crypt-villus architecture and goblet cell functionality. Additionally, as a commensal probiotic, LRa05 could ferment dietary fibers to produce SCFAs (e.g., acetate, propionate, butyrate), which exert pleiotropic restorative effects [31,32]. The histomorphometric data demonstrated that *Lactobacillus rhamnosus* LRa05 mitigated constipation-associated colonic degeneration through mucosal barrier regeneration and anti-inflammatory reprogramming. These structural improvements likely synergize with its prokinetic effects (Figure 1a) and serotonergic modulation (Table 2), collectively restoring gut homeostasis.

### 3.4. LRa05’s Role in Affecting the Content of SCFAs

Short-chain fatty acids (SCFAs), the primary metabolites of gut microbial fermentation, alleviate constipation through multiple mechanisms: modulating gut microbiota composition, stimulating luminal hydration, and preserving intestinal barrier integrity [33].

As shown in Table 2, constipated mice treated with loperamide in the MC group exhibited significantly reduced cecal SCFA levels relative to the NC group (*p* < 0.05). The levels of acetic acid (9.46 µmol/g), propionic acid (5.43 µmol/g), butyric acid (4.17 µmol/g), isobutyric acid (3.78 µmol/g), valeric acid (1.32 µmol/g), and isovaleric acid (2.10 µmol/g) were notably lower in the NC group. These reductions could be attributed to constipation-induced fecal stasis disrupting the gut microbial ecosystem, leading to the suppression of SCFA-producing taxa, such as *Bacteroides*, *Clostridium*, and *Roseburia*. This systemic SCFA depletion is likely to contribute to clinical symptoms, like motility impairment and stool dryness, as SCFAs are critical energy sources for colonocytes.

Notably, *Lactobacillus rhamnosus* LRa05 intervention dose-dependently restored SCFA production, with the high-dose group (LRa05.H) showing superior efficacy (*p* < 0.01). Specifically, the high-dose LRa05 could increase the acetic acid and butyric acid contents to 38.2 µmol/g and 12.3 µmol/g, respectively, which were higher than those in the NC group (27.1 µmol/g and 8.55 µmol/g). Acetic acid activates G protein-coupled receptor 43 (GPR43) to trigger motility-stimulating peptides (e.g., PYY, GLP-1), aligning with clinical hormone-regulatory strategies, while butyrate reduces inflammation via histone deacetylase (HDAC) inhibition, offering a “dual-action probiotic” for clinical constipation [34]. This dual action mitigates “leaky gut”-induced inflammation and preserves ENS and ICC function critical for motility [35]. Propionate enhances vagal afferent signaling to the brainstem, thereby stimulating defecation reflexes. Valerate modulates serotonin receptor (5-HT4) sensitivity, potentiating smooth muscle contraction. Mechanistically, LRa05 inhibits pathogens (e.g., *Escherichia*, *Enterobacter*) and promotes SCFA-producing microbiota, reactivating fiber fermentation. It upregulates enzymes like butyryl-CoA transferase (for butyrate) and phosphate acetyltransferase (for acetate) restoring SCFA pathways. Butyrate-induced mucin synthesis in humans supports LRa05’s potential to reduce stool hardness [36].

These results suggest LRa05 could relieve constipation via dual pathways: SCFA-mediated motility enhancement and butyrate-driven barrier protection. Compared to the positive control (mosapride), LRa05 showed stronger SCFA restoration, highlighting microbiota-targeted therapies as a promising approach. For instance, acetic acid levels in the mosapride group reached 21.6 µmol/g, compared to 38.1 µmol/g in LRa05.H and 9.46 µmol/g in MC (Table 2). This disparity suggested that mosapride citrate and LRa05 operate through divergent mechanistic pathways to alleviate constipation. The differential efficacy between mosapride citrate and LRa05 underscores the importance of microbiota-centric therapies in addressing SCFA-mediated constipation. While mosapride transiently improves motility, probiotics achieve sustained recovery by rebuilding the gut ecosystem and enhancing host-microbe cross-talk. Combining prokinetics with probiotics may synergize motility enhancement and SCFA restoration. Notably, while SCFA levels correlated with symptom improvement after LRa05 treatment, causal links remain unproven. Functional studies (SCFA supplementation or receptor blockade) are needed to confirm their role in constipation relief.

### 3.5. LRa05’s Role in Affecting the Expression of Linked Genes

To delve deeper into the mechanism underlying the effects of *Lactobacillus rhamnosus* LRa05, the critical genes governing gastrointestinal motility and fluid homeostasis were profiled systematically (Figure 2). ICCs, the pacemakers of gastrointestinal rhythmicity, coordinate smooth muscle contractions through electrical slow waves [37,38]. The SCF/C-Kit axis is indispensable for ICC maintenance, with SCF binding to its tyrosine kinase receptor C-Kit to drive ICC proliferation and network formation [39]. The dysregulation of this pathway disrupts slow-wave generation, precipitating motility disorders [40].

The qRT-PCR data of mRNA levels revealed constipation-induced colonic suppression of both SCF and C-Kit (4.41- and 3.8-fold downregulation, respectively) in the MC group versus the NC group (*p* < 0.01 or *p* < 0.001; Figure 2a,b). This was because SCF/C-Kit signaling was essential for the growth, survival, and functional maintenance of ICC—the pacemaker cells responsible for generating intestinal slow-wave potentials. Reduced SCF expression and C-Kit receptor downregulation disrupt ICC network integrity, leading to diminished slow-wave activity and impaired gastrointestinal motility. This molecular perturbation aligns with the pathophysiological hallmark of chronic constipation—the loss of ICC density and function. ICC depletion directly compromises the intrinsic myoelectric activity of the intestinal smooth muscle, resulting in aberrant peristaltic contractions and reduced propulsive motility, which are cardinal features of hypomotility disorders. The SCF and C-Kit axis dysfunction thus represent a critical molecular node linking colonic mucosal signaling to motility deficits in constipation, underscoring its role as a potential target for restoring ICC-mediated gastrointestinal rhythm.

Following treatment with various doses of LRa05 or the administration of mosapride citrate tablets, all experimental groups demonstrated differential recovery responses. LRa05 intervention dose-dependently restored SCF and C-Kit expression levels (*p* < 0.01 for LRa05.L and *p* < 0.001 for LRa05.H vs. MC), showing 73.1% and 47.2% recovery with low-dose treatment, and 92.6% and 69.6% recovery with high-dose treatment. Mechanistically, SCF activation bound to C-Kit receptors on ICC progenitor cells, promoting their proliferation and differentiation. This molecular restoration correlated with the histological observations of muscularis propria thickening (Figure 1c) and enhanced intestinal propulsion (Figure 1a), suggesting LRa05 reactivates ICC-dependent pacemaking to reinstate peristaltic rhythms. This recovery likely stemmed from LRa05-mediated GBA regulation. Probiotic SCFAs boosted enterochromaffin cell activity, increasing 5-HT biosynthesis (Table 2). 5-HT activates 5-HT4 receptors on epithelial cells or ICCs, triggering cAMP/PKA signaling that enhances SCF secretion [41]. SCF, in turn, binds to C-Kit on ICCs, promoting their proliferation and restoring slow-wave generation (Figure 2b). In addition, Lactobacillus strains, including LRa05, could directly engage Toll-like receptor 2 (TLR2) on intestinal epithelial cells, activating MyD88-dependent signaling to upregulate SCF transcription [42]. SCFAs amplify this process by stabilizing epithelial barrier function, ensuring sustained SCF release [27]. This suggested LRa05 may regenerate ICC networks—a novel biological strategy for clinical ICC-targeted therapy.

The PC group (mosapride citrate-treated) exhibited only a partial restoration of SCF and C-Kit expression (e.g., 53.8% and 32.4% recovery vs. MC, respectively), significantly lagging behind the probiotic intervention groups. This disparity implies that mosapride’s monotherapy primarily targets acute motility enhancement rather than addressing the multi-factorial pathophysiology of constipation. This differential therapeutic effect can be mechanistically attributed to Mosapride’s role as a selective 5-HT4 receptor agonist, which transiently enhances serotonin-dependent smooth muscle contractility. However, it lacks direct regulatory effects on ICC progenitor cell proliferation or SCF transcriptional activity-key processes required for sustainable ICC network regeneration. Moreover, Mosapride fails to restore intestinal microbiota diversity or SCFA concentrations (notably butyrate), which are critical for epithelial SCF secretion and ICC metabolic homeostasis.

Complementary to motor function, the levels of aquaporins (AQP4 and AQP8) governing intestinal water transport were assessed. Compared with the NC group, chronic constipation led to a significant upregulation of AQP4 and AQP8 in the MC group, as shown in Figure 2c,d. Specifically, the expression level of AQP4 was 1.68-fold higher (*p* < 0.05), and that of AQP8 was 1.83-fold higher (*p* < 0.01) than those in the NC group. This aligned with prior evidence that AQP overexpression exacerbates pathological fluid retention and stool desiccation [35]. LRa05 administration reversed this dysregulation, with high-dose treatment reducing AQP4 and AQP8 expression to 47.3% and 38.1% of MC levels (*p* < 0.01), effectively rebalancing luminal hydration. The overexpression of AQPs in constipation is often driven by inflammatory cytokines (e.g., TNF-α, IL-6) that upregulate AQP transcription via NF-κB/MAPK pathways. LRa05-derived SCFAs could suppress these pathways, reducing AQP4/AQP8 expression [43,44]. SCFAs could interact with GPR43 receptors on enterocytes, activating intracellular calcium signaling to modulate AQP trafficking or degradation, thereby reducing membrane AQP density [22]. From a gastrointestinal peptide synergy perspective, *Lactobacillus rhamnosus* LRa05 lowers inhibitory peptides (VIP, SS; Table 1), which were known regulators of AQP activity. Reduced VIP/SS levels could decrease cAMP signaling in enterocytes, thereby indirectly normalizing AQP-mediated water reabsorption. Concurrently, elevated 5-hydroxytryptamine (5-HT; Table 2) and restored MTL enhance fluid secretion by stimulating chloride channels (e.g., cystic fibrosis transmembrane conductance regulator, CFTR), counteracting AQP-driven water retention. The coordinated actions of AQP modulation and neuropeptide regulation synergistically maintain optimal fecal viscoelasticity (rheological plasticity) and prevent stool dehydration, thereby preserving intestinal transit efficiency.

This dual modulation of inhibitory–excitatory peptide networks likely rebalanced luminal hydration without the electrolyte risks of traditional drugs—matching clinical goals of preventing excessive water reabsorption. At the motility level, SCF/C-Kit activation rebuilds ICC networks, rescuing slow-wave activity for propulsive contractions [45]. Concurrently, at the osmoregulatory level, AQP suppression prevents excessive water extraction, maintaining stool plasticity [46]. This mechanistic synergy could explain the accelerated fecal transit, improved stool hydration, and normalized defecation frequency observed in LRa05-treated groups (Figure 1a).

### 3.6. Functional Role of LRa05 in Modulating the Gut Microbiota

To evaluate the microbiota-modulatory effects of *Lactobacillus rhamnosus* LRa05 in constipated mice, we conducted 16S rRNA gene sequencing on colonic contents collected at the study endpoint. Alpha diversity metrics (Chao1 and observed OTUs for richness, Shannon and Simpson for diversity) were analyzed via species richness and community diversity indices (Figure 3a). Beta diversity was assessed using PCoA and NMDS based on Bray–Curtis dissimilarity (Figure 3b).

In the MC group, significant (*p* < 0.05) reductions were observed in both microbial richness (Chao1 and observed OTUs) and diversity (Shannon and Simpson indices) compared to the NC group. The LRa05 intervention reversed these perturbations in a dose-dependent manner. The high-dose LRa05.H group showed significantly higher Chao1 richness, observed OTUs (*p* < 0.001), and Shannon and Simpson indices (*p* < 0.01) than the low-dose LRa05.L and PC groups (Figure 3a). PCoA and NMDS revealed distinct clustering patterns. The MC group separated from the NC group along PC2 (19.1% variance), indicating constipation-induced gut microbiota dysbiosis (Figure 3b). Notably, both LRa05.H and LRa05.L groups overlapped with the NC group, indicating a partial restoration of microbiota composition. Conversely, the PC group showed a clear separation along the first principal component (PC1, 44.0% variance) relative to other groups, highlighting the distinct role of moxapride (used in the PC group) in modulating gut microbiota composition.

Consistent with established gut microbiota profiles, Bacteroidetes and Firmicutes dominated (>90% combined abundance) at the phylum level (Figure 4a) [47]. Loperamide-induced constipation disrupted gut microbiota composition, with Bacteroidetes abundance decreasing by 21.2% and Firmicutes increasing by 26.1% relative to the NC control (*p* < 0.01). This microbial dysbiosis resulted in a significantly reduced Firmicutes/Bacteroidetes (F/B) ratio in the MC group (1.01 ± 0.12) compared to the NC group (1.62 ± 0.18; *p* < 0.05). LRa05 intervention restored the F/B ratio in a dose-dependent manner (LRa05.L: 1.38 ± 0.15; LRa05.H: 1.52 ± 0.14), with the high-dose group achieving NC-equivalent levels (*p* > 0.05). This finding aligns with previous studies demonstrating that lactic acid bacteria stabilize F/B ratios to mitigate inflammation and enhance epithelial barrier function [48].

Additionally, constipated microbiota are characterized by decreased beneficial bacteria and increased potential pathogens [49]. At the genus level (Figure 4b and Appendix A), loperamide induced significant compositional shifts. LRa05 intervention notably restored beneficial taxa, including Parabacteroides, *Lachnospiraceae*_NK4A136_group, *Oscillibacter*, *Alloprevotella*, and *Rikenellaceae*_RC9_gut_group, aligning with prior studies. For instance, functional fruit drinks relieved constipation by upregulating *Alloprevotella* and enhancing barrier function [50], while *Lactobacillus* salivus Li01 restored beneficial bacteria, like *Lachnospiraceae*_NK4A136_group and *Rikenellaceae*_RC9_gut_group abundances to reduce inflammation [51]. Emerging evidence indicates Parabacteroides and *Oscillibacter* convert dietary tryptophan into indole-3-propionic acid (IPA). This compound activates the aryl hydrocarbon receptor (AhR), which reduces brain inflammation and supports gut nervous system health. Mechanistically, *Alloprevotella* negatively correlates with pro-inflammatory cytokines and produces SCFAs, like butyrate, to mitigate intestinal inflammation [52]. *Lachnospiraceae*_NK4A136_group, an anaerobic spore-forming probiotic, ferments plant polysaccharides into SCFAs to protect intestinal mucosa and inversely correlates with inflammation markers. *Rikenellaceae*_RC9_gut_group mediates anti-inflammatory effects through immune modulation, strengthening intestinal barrier integrity. In summary, *Lactobacillus rhamnosus* LRa05 exerted functional effects by restoring beneficial bacterial genera, promoting SCFA production, modulating inflammatory pathways, and enhancing intestinal barrier function. These coordinated effects are underabundant in clinical constipation and correlate with reduced inflammation and improved motility [53].

The heatmap and LEfSe analysis (LDA > 3.0, Figure 5a,b) identified distinct microbial signatures across groups. Compared with the other group, the MC group exhibited a significant enrichment of *Desulfovibrio*, *Candidatus_Saccharimonas*, *Enterorhabdus*, and *Rikenella* (LEfSe analysis, LDA score > 3.0, *p* < 0.05). These microbial taxa are linked to intestinal mucosal barrier dysfunction (e.g., leaky gut, lower tight junction proteins) and higher inflammation (IL-6, TNF-α, IL-1β) in animals and humans [54]. Notably, *Enterorhabdus* has been specifically implicated in triggering NF-κB-mediated inflammatory cascades through its metabolite interactions with intestinal epithelial cells, as demonstrated by in vitro transwell co-culture systems [55]. It is worth noting that these inflammatory cytokines or signaling pathways may exacerbate constipation symptoms. For instance, IL-6 slows intestinal peristalsis by inhibiting cholinergic neuron activity. TNF-α induces segmental hyperkinesia and transmission delay via the RhoA/ROCK-mediated enhancement of smooth muscle contraction persistence, while IL-1β exacerbates fecal dryness through downregulating intestinal epithelial aquaporin and reducing luminal water secretion. This finding was further validated in a correlation analysis (Figure 5c), where *Enterorhabdus* and *Rikenella* demonstrated significant positive correlations with AQP4 and AQP8, respectively.

The LRa05.H group restored ecological equilibrium via the enrichment of bacteria with SCFA synthesis, or immune regulatory functions (such as Bacteroides, Parabacteroides, *Paraprevotella*, *Colidextribacter*, *Rikenellaceae*_RC9_gut_group) [20]. For instance, *Bacteroides* and *Parabacteroides* species predominantly mediate SCFA production and mucin degradation, creating a prebiotic niche for secondary fermenters. These taxa provide metabolic substrates (e.g., acetate, propionate) and mucin-derived oligosaccharides that sustain microbial communities critical for intestinal homeostasis [56]. The LRa05.L group boosted beneficial bacteria, such as *Eubacterium*_*xylanophilum*_group, *Oscillibacter*, *Alistipes*, *Odoribacter*, and *Lachnospiraceae*_NK4A136_group. These taxa support gut barrier integrity, neurotransmitter production and mental health, immune regulation and inflammation reduction, lipid metabolism, and SCFA production for colon health [57]. Notably, the preferential proliferation of *Oscillibacter* and *Alistipes* exerts synergistic neuroprotective and immunomodulatory effects [58]. This suggested LRa05 may improve clinical constipation via microbiota–metabolism–immunity axis remodeling—providing a probiotic candidate for microbiota-targeted therapy.

The LRa05.L (low-dose) and LRa05.H (high-dose) groups exhibited distinct induction patterns of beneficial taxa, demonstrating a dose-dependent probiotic modulation of gut microbial ecology. Additionally, the PC group exhibited a unique gut microbiota modulation profile distinct from both LRa05.L and LRa05.H probiotic interventions. Specifically, *Akkermansia*, *Alloprevotella*, *Prevotellaceae*_*UCG-001*, and *Muribaculum* were significantly enriched in the PC group compared to probiotic-treated cohorts, with *Alloprevotella* demonstrating the most pronounced enrichment. The PC-induced microbiota profile reflected the ecological mechanisms favoring mucin- and plant glycan-adapted taxa rather than functional resilience, distinct from the probiotic cross-feeding networks observed in LRa05 groups [59].

The microbial community differentiation was further supported by the correlation analysis (Figure 5c). The MC group exhibited significant microbial–metabolic correlations distinct from the probiotic-treated LRa05.H group. The *Desulfovibrio* and *Candidatus_Saccharimonas* with higher abundances in the MC group were significantly negatively correlated with fecal water content, propulsion rate, levels of gut motility activators (GAS, MTL, SP, 5-HT), and the expression of SCF/C-kit. In contrast, they had significant positive correlations with time to first black feces and levels of motility inhibitors (SS, VIP, ET-1). Notably, *Rikenella*—another constipation-associated genus enriched in the MC group—demonstrated a notable negative correlation with SCFA production (butyrate: r = −0.65; acetate: r = −0.61; *p* < 0.05). This aligned with its known role in degrading mucin glycans and releasing pro-inflammatory mediators (e.g., IL-6, TNF-α), which suppress colonic fermentation and epithelial energy metabolism. The dominance of *Desulfovibrio* in the MC group likely impairs gut motility through hydrogen sulfide-mediated smooth muscle relaxation, while its mucolytic activity exacerbates intestinal barrier dysfunction, triggering the upregulation of SS/VIP [60]. Concurrently, *Rikenella* thrives in dysbiotic, inflamed environments by preferentially metabolizing host glycans over dietary fibers. This metabolic shift diverts microbial activity toward lactate and ammonia production, further acidifying the intestinal lumen and inhibiting SCFA biosynthesis. Collectively, these distinct microbial–metabolic interactions in the MC group highlight constipation-associated dysbiosis disrupts. This disruption could maintain gut motility issues and trigger inflammatory responses [61].

In contrast, probiotic-enriched genera *Paraprevotella* and *Parabacteroides* in the LRa05.H group exhibited inverse correlations with these motility and metabolic parameters, suggesting beneficial modulation of gut function. Specifically, *Parabacteroides* likely enhance motility via succinate-TGR5 receptor signaling, which potentiates 5-HT biosynthesis in enterochromaffin cells. Additionally, *Paraprevotella* could produce anti-inflammatory 3-indolepropionic acid to suppress Endothelin-1 through NF-κB (Nuclear Factor kappa-light-chain-enhancer of activated B cells) inhibition [62]. Furthermore, genera Eubacterium_xylanophilum_group, *Colidextribacter*, and *Rikenellaceae_RC9*_gut_group demonstrated significant positive correlations with SCFA production, underscoring their roles in metabolic regulation. Notably, taxa enriched in the positive control PC group (*Akkermansia*, *Alloprevotella*, *Prevotellaceae*) exhibited negligible significant correlations with these functional parameters. The above findings suggested that dietary probiotic supplementation alleviates constipation symptoms through a multi-tiered microbiota–host signaling axis. For example, this could involve SCFA biosynthesis, gastrointestinal regulatory peptide modulation, SCF/C-Kit pathway reactivation, and AQP homeostasis restoration.

To further verify these correlations, we performed PICRUSt2 analysis on the 16S rRNA sequencing data to infer microbial functional pathways (see Appendix A). The results showed that the enriched taxa in the LRa05.L and LRa05.H probiotic interventions groups exhibited significant enrichment in multiple functional pathways related to SCFA biosynthesis, amino acid metabolism, energy metabolism, and signaling cascades. For example, pyruvate metabolism and butanoate metabolism serve as core pathways for the biosynthesis of acetic acid, propionic acid, and butyric acid [63]. The enriched SCFA biosynthesis and amino acid metabolic pathways could provide amino acid raw materials or energy for AQP synthesis. Bacteria sense osmotic stress or proteins through two-component systems, directly activating the expression of AQP proteins [64]. Importantly, these associations are correlative, and future studies using gnotobiotic models or in vitro cultures are needed to validate direct mechanistic roles in symptom relief.

The multi-targeted effects of LRa05 differentiate it from conventional laxatives or prokinetic agents by addressing both symptomatic and mechanistic drivers of gastrointestinal dysfunction (Figure 6). Unlike traditional laxatives (such as osmotic or stimulant types) that primarily enhance bowel movement through passive water retention or direct intestinal irritation, LRa05 engages in the active modulation of molecular pathways (such as SCF/c-Kit, AQP4/AQP8) and microbial-metabolic balance (such as SCFA production, beneficial bacteria enrichment). This dual focus on restoring physiological functions (such as tight junction integrity, neurotransmitter homeostasis) rather than merely alleviating symptoms may reduce reliance on repeated dosing and lower recurrence rates, as it targets the root causes of dysmotility and microbial imbalance.

Conventional prokinetic agents (for instance metoclopramide) often exhibit narrow specificity for neurotransmitter pathways (for instancedopamine or serotonin receptors) and carry risks of adverse events, such as arrhythmias, extrapyramidal symptoms, or tolerance development. In contrast, LRa05’s broad regulatory effects on multiple signaling molecules (MTL, GAS, SP, 5-HT, SS, ET-1, VIP) may achieve comparable motility enhancement through a more physiologically integrated mechanism, potentially minimizing off-target effects. Additionally, by promoting microbial diversity and suppressing harmful species, LRa05 supports long-term gut health, a benefit not typically associated with short-acting pharmaceutical agents that do not address microbial dysregulation.

While this study demonstrates LRa05’s multi-targeted efficacy in a murine constipation model, key limitations include the model’s inability to fully mimic complex human chronic constipation and species-specific differences in gut microbiota and physiology. Translational research is needed to validate LRa05’s long-term efficacy and safety in humans, including dose scaling and an assessment of individual responses based on baseline microbiota profiles. The dedicated limitations section notes that human clinical trials are essential to evaluate its potential for reducing recurrence and addressing translational gaps.

## 4. Conclusions

This mechanistic framework provides a basis for targeted constipation interventions (differentiating between dehydration- and motility-related subtypes) and highlights the unique advantages of probiotic multi-target intervention. Unlike prior single-target studies (focused on microbiota or SCFAs), our findings suggest *Lactobacillus rhamnosus* LRa05 acts as a potential multi-target probiotic for constipation management. The LRa05 intervention improved gastrointestinal transit, as shown by a shorter time to first black feces and higher fecal moisture. The treatment also regulated gut hormones, increasing motilin while decreasing vasoactive intestinal peptide to restore enteroendocrine balance. At the molecular level, LRa05 normalized signaling by upregulating Cajal interstitial cell markers and suppressing AQP4/AQP8 overexpression. Additionally, it remodeled the gut microbiota, boosting SCFA production and enriching beneficial microbes, like *Parabacteroides* and *Lachnospiraceae*, which differed from antibiotic-treated controls. These findings highlight the probiotic’s dual role in mediating microbiota–gut barrier cross-talk and regulating neuroendocrine axes via integrated ICC–hormonal–microbial signaling. Mechanistically, LRa05 demonstrates translational potential for constipation management by simultaneously addressing motility dysfunction, mucosal injury, and luminal dehydration. Notably, this probiotic strain selectively enriched anti-inflammatory and SCFA-producing genera (*Parabacteroides*, *Lachnospiraceae*), with microbial profiles exhibiting a marked divergence from the antibiotic-treated control group. Such ecological specificity highlights the functional advantages of probiotic interventions over broad-spectrum antimicrobials. Notably, these mechanistic inferences require validation. While our findings link LRa05 to microbiota remodeling and SCFA elevation, causal relationships between these effects and symptom improvement remain unproven. Functional studies (e.g., SCFA supplementation or receptor blockade) are needed to disentangle direct mechanisms, and translational steps—such as human trials for safety and dose optimization—are critical to confirm clinical efficacy.

## Figures and Tables

**Figure 1 foods-14-02293-f001:**
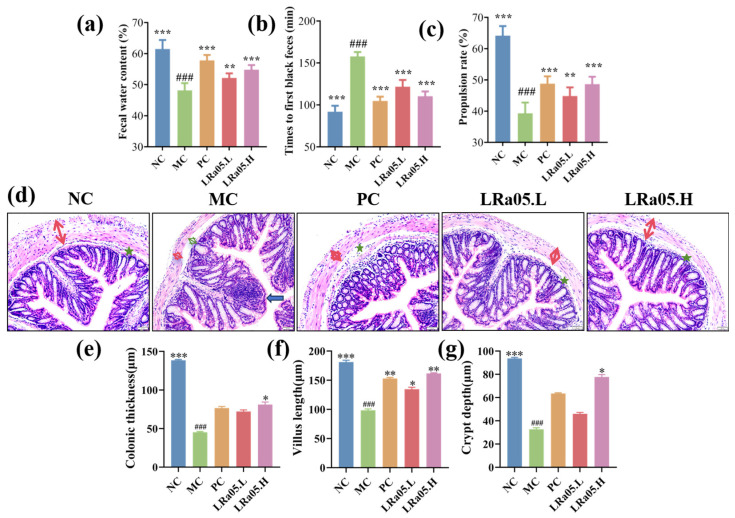
Effects of *Lactobacillus rhamnosus* LRa05 on constipation-related parameters and intestinal tissue damage. (**a**) Fecal water content (% dry weight); (**b**) colonic transit time (min); (**c**) intestinal propulsion rate (%); (**d**) HE-stained images of mice colon tissue. The red arrow represents intestinal wall thickness. The green star represents mucosal atrophy with muscularis dissociation. The blue arrow represents focal inflammatory infiltrates. (**e**) thickness of the colon wall (μm); (**f**) crypt depth (μm); (**g**) villus height (μm). Data are presented as mean ± standard error of the mean (SEM) with a sample size of *n* = 10. For statistical significance, the symbol ### denotes *p* < 0.001 (vs. NC group); *** representing *p* < 0.05, ** representing *p* < 0.01, and *** representing *p* < 0.001 (vs. MC group).

**Figure 2 foods-14-02293-f002:**
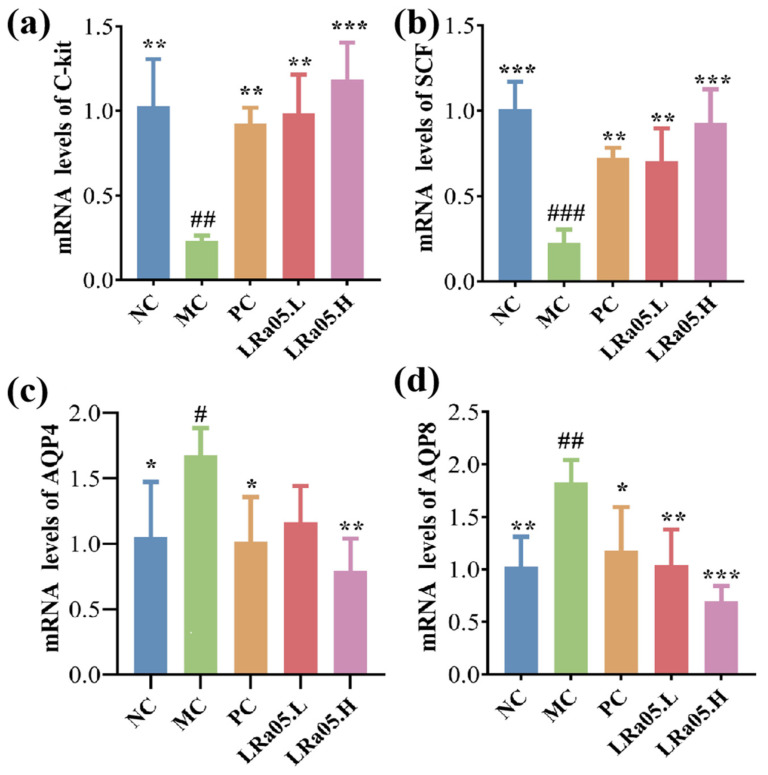
Transcriptional regulation of stem cell factor (SCF, (**a**)), Tyrosine Protein Kinase Receptor Kit (C-Kit, (**b**)), aquaporin 4 (AQP4, (**c**)), and aquaporin 8 (AQP8, (**d**)) in the colons of mice. Data are presented as mean ± standard error of the mean (SEM) with a sample size of *n* = 10. For statistical significance, the symbol # denotes *p* < 0.05, ## denotes *p* < 0.01, ### denotes *p* < 0.001 (vs. NC group); * representing *p* < 0.05, ** representing *p* < 0.01, and *** representing *p* < 0.001 (vs. MC group).

**Figure 3 foods-14-02293-f003:**
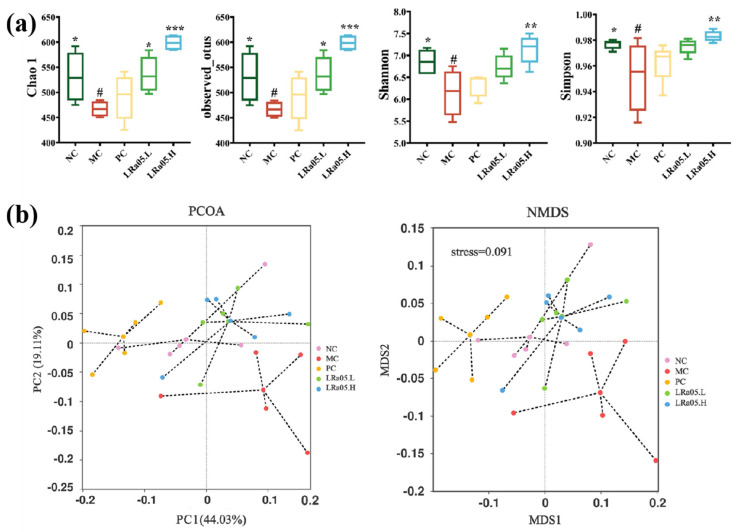
LRa05-mediated modulation of species diversity and differentiation in the gut microbiota. (**a**) Alpha diversity. (**b**) Beta diversity. Data are presented as mean ± standard error of the mean (SEM) with a sample size of *n* = 10. For statistical significance, the symbol # denotes *p* < 0.05 (vs. NC group); * representing *p* < 0.05, ** representing *p* < 0.01, and *** representing *p* < 0.001 (vs. MC group).

**Figure 4 foods-14-02293-f004:**
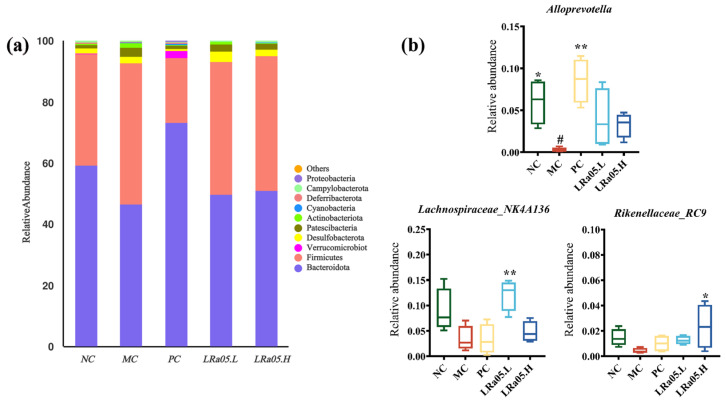
Modulation of gut microbiota composition by LRa05 intervention. (**a**) Phylum-level distribution of intestinal microbiota. (**b**) Relative abundances of selected bacterial genera demonstrating significant alterations. Data are presented as mean ± standard error of the mean (SEM) with a sample size of *n* = 10. For statistical significance, the symbol # denotes *p* < 0.05 (vs. NC group); * representing *p* < 0.05, ** representing *p* < 0.01 (vs. MC group).

**Figure 5 foods-14-02293-f005:**
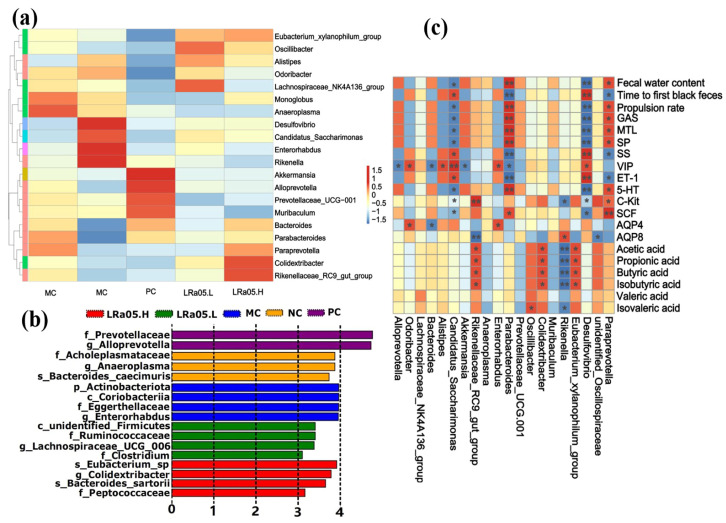
Microbial biomarkers and functional correlations. (**a**) Hierarchical clustering was visualized using a heatmap. (**b**) Linear discriminant analysis (LDA) effect size (LEfSe) (LDA score ≥ 3.0). (**c**) Spearman correlation heatmap depicting genus-level associations with constipation-related phenotypes. r = 0.7–1.0: dark reds. r = 0.3–0.7: light red. r = −0.3–0.3: white. r = −0.7–−0.3: light blue. r = −1.0–−0.7: dark blue. Significant correlations are marked with * *p* < 0.05 and ** *p* < 0.01. Color intensity reflects correlation strength (blue = negative, red = positive), with significant correlations denoted by * *p* < 0.05 and ** *p* < 0.01.

**Figure 6 foods-14-02293-f006:**
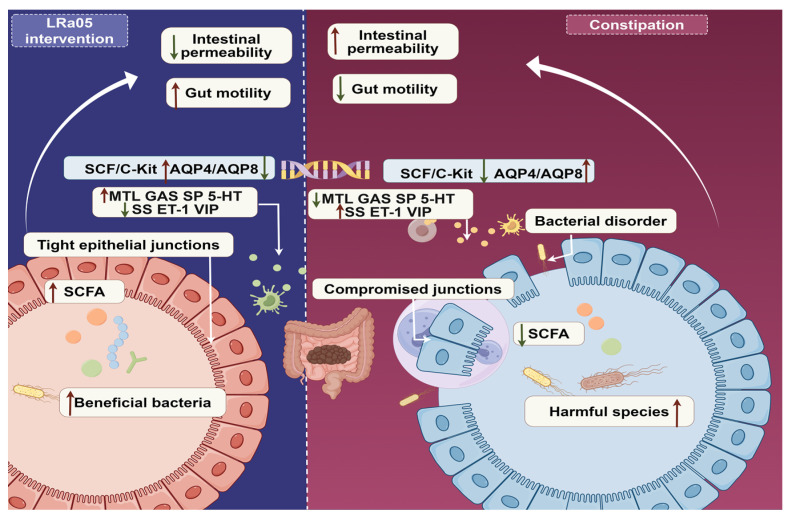
Schematic diagram of LRa05’s mechanisms against constipation. The red arrow indicates enhancement. The green arrow indicates attenuation.

**Table 1 foods-14-02293-t001:** Genes and primers sequences used for qRT-PCR analysis.

Gene	Primer
AQP4	Forward: CTTTCTGGAAGGCAGTCTCAG Reverse: CCACACCGAGCAAAACAAAGAT
AQP8	Forward: AGATGCCGTGTGTTCTGGTA Reverse: AGTGTCCACCGCTGATGTTC
C-Kit	Forward: GAGTGTAAGGCCTCCAACGA Reverse: GGGCCTGGATTTGCTCTTTGT
SCF	Forward: TCAGGGACTACGCTGCGAAAG Reverse: AAGAGCTGGCAGACCGACTCA
β-actin	Forward: GTGACGTTGACATCCGTAAAGA Reverse: GCCGGACTCATCGTACTCC

**Table 2 foods-14-02293-t002:** The serum concentrations of gastrointestinal regulatory peptides and the contents of SCFAs in various groups of mice.

Index	Group				
	NC	MC	PC	LRa05.L	LRa05.H
**S** **erum concentrations of gastrointestinal regulatory peptides**					
MTL	395.05 ± 8.57 ^a^	264.75 ± 13.50 ^b^	376.81 ± 10.92 ^a^	275.43 ± 16.47 ^b^	300.24 ± 14.22 ^b^
GAS	66.36 ± 2.24 ^a^	36.79 ± 2.35 ^c^	64.43 ± 2.11 ^a^	43.72 ± 2.14 ^bc^	48.24 ± 1.55 ^b^
SP	304.12 ± 7.99 ^a^	176.36 ± 8.54 ^c^	280.57 ± 12.45 ^a^	183.89 ± 10.75 ^bc^	223.52 ± 5.64 ^b^
SS	113.06 ± 3.34 ^c^	174.10 ± 5.68 ^a^	103.09 ± 3.95 ^c^	160.71 ± 6.69 ^ab^	142.66 ± 7.07 ^b^
ET-1	60.91 ± 1.92 ^d^	135.83 ± 5.47 ^a^	68.52 ± 1.10 ^d^	110.82 ± 3.78 ^b^	86.33 ± 2.51 ^c^
VIP	120.01 ± 5.93 ^c^	216.68 ± 10.02 ^a^	108.45 ± 8.53 ^c^	182.68 ± 12.12 ^b^	133.70 ± 7.87 ^c^
5-HT	5.68 ± 0.20 ^d^	2.24 ± 0.06 ^a^	5.06 ± 0.16 ^d^	2.90 ± 0.08 ^b^	3.66 ± 0.11 ^c^
**SCFA contents**					
Acetic acid	27.11 ± 1.99 ^bc^	9.46 ± 0.91 ^d^	21.64 ± 1.33 ^c^	26.68 ± 0.51 ^b^	38.15 ± 1.13 ^a^
Propionic acid	9.25 ± 0.65 ^ab^	5.43 ± 0.23 ^d^	7.77 ± 0.65 ^c^	9.59 ± 0.72 ^ab^	10.75 ± 0.49 ^a^
Butyric acid	8.55 ± 0.50 ^bc^	4.17 ± 0.27 ^d^	7.03 ± 0.50 ^c^	9.61 ± 1.22 ^b^	12.34 ± 0.58 ^a^
Isobutyric acid	4.20 ± 0.10 ^a^	3.78 ± 0.04 ^b^	4.01 ± 0.07 ^ab^	4.21 ± 0.08 ^a^	4.23 ± 0.10 ^a^
Valeric acid	1.83 ± 0.10 ^a^	1.32 ± 0.05 ^b^	1.60 ± 0.09 ^ab^	1.98 ± 0.14 ^a^	1.95 ± 0.17 ^a^
Isovaleric acid	2.53 ± 0.11 ^a^	2.10 ± 0.05 ^b^	2.32 ± 0.08 ^ab^	2.58 ± 0.12 ^a^	2.52 ± 0.12 ^a^

Data represent the mean ± SEM (*n* = 10). Different letters within the same row denote significant differences among distinct groups, with a significance level of *p* < 0.05.

## Data Availability

The original contributions presented in this study are included in the article/Appendix A. Further inquiries can be directed to the corresponding authors.

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
