# Peer review of "Lactobacillus rhamnosus LRa05 Alleviates Constipation via Triaxial Modulation of Gut Motility, Microbiota Dynamics, and SCFA Metabolism"

_foods, 2025, doi:10.3390/foods14132293_

Round 1
Reviewer 1 Report
Comments and Suggestions for Authors
This manuscript presents a thorough and well-structured investigation into the effects of Lactobacillus rhamnosus LRa05 on loperamide-induced constipation in mice. The authors employ a comprehensive suite of methods—ranging from behavioral and physiological assessments to molecular, metabolomic, and microbiome analyses. The study is timely and relevant, especially given the limitations of current constipation therapies and the growing interest in multi-target probiotics.
Strengths
Comprehensive Mechanistic Approach:
The integration of gut motility, regulatory peptides, aquaporin expression, SCFA quantification, and microbiota profiling provides a robust mechanistic framework. This multi-layered analysis is a major strength.
Clinical Relevance:
The outcomes measured—such as stool frequency, stool consistency (fecal water content), intestinal transit time, and restoration of key microbiota—are directly translatable to clinical endpoints in constipation management. The inclusion of both low and high probiotic doses, as well as a positive control, strengthens the translational value.
Multi-Targeted Efficacy:
The demonstration that LRa05 can simultaneously modulate gut motility, mucosal barrier function, and microbiota composition is a notable advance over single-mechanism interventions.
Areas for Improvement
- Methods Transparency and Reproducibility
- Sample Size Justification:
Please clarify whether a power analysis was performed to determine the group size (n=10). If not, provide a rationale or reference supporting the chosen sample size. - Randomization and Blinding:
The manuscript mentions stratified randomization but does not specify blinding procedures. Please detail how animals were randomized and whether outcome assessors were blinded to group assignments. - Primer and Reagent Details:
While primer sequences are provided in Supplementary Table S1, including them in the main text or as a dedicated table would improve reproducibility. - Statistical Methods:
Specify which statistical tests were applied to each dataset, how normality and variance were assessed, and whether corrections for multiple comparisons were used. - Data Presentation
- Figure Clarity:
Some figures (e.g., microbiota heatmaps, Figure 4b) are visually dense and difficult to interpret. Highlight the most clinically relevant genera and consider moving detailed data to supplementary figures. - Statistical Annotation:
Ensure all statistical symbols (e.g., *, ***, #) are clearly defined in each figure legend. Expand legends to provide context for group comparisons. - Supplementary Data:
Move raw or highly detailed data (e.g., GC-MS chromatograms, ELISA outputs) to supplementary materials to streamline the main figures. - Clinical Translation and Impact
- Explicit Clinical Correlation:
For each major result, briefly state its clinical implication. For example, relate improvements in fecal water content and transit time to symptom relief in patients (softer stools, more regular bowel movements). - Comparison with Standard Therapies:
Discuss how LRa05’s multi-targeted effects compare to conventional laxatives or prokinetic agents, especially regarding recurrence rates and adverse events. - Translational Steps:
Recommend future research directions, such as human clinical trials, dose-finding, and long-term safety studies, to bridge preclinical findings to clinical practice. - Interpretation and Framing
- Mechanistic Claims:
Avoid overextending mechanistic interpretations, particularly regarding gut-brain axis effects, unless supported by direct evidence. - Limitations:
Explicitly discuss the limitations of the murine model and the need for human validation. - English Language and Style
- Revise for conciseness and readability. Shorten lengthy sentences and replace technical jargon with more accessible language where possible.
- Data and Code Availability
- For enhanced reproducibility, consider depositing raw data (e.g., 16S rRNA sequences, GC-MS data, qPCR Ct values) in a public repository and provide accession numbers.
---------------------------------------------------------------------------
Specific Suggestions
Introduction:
Streamline background information and focus on the unique clinical and mechanistic aspects of LRa05.
Methods:
Add more detail on animal welfare measures, exclusion criteria, and statistical analysis steps.
Results:
Present fecal water content, transit time, and propulsion rate in separate, clearly labeled panels. In microbiota analyses, highlight functionally relevant taxa.
Discussion:
Compare the magnitude of effect to benchmarks from clinical studies and discuss the potential for LRa05 to fill unmet needs in constipation therapy.
Figures:
Improve resolution, unify font sizes, and ensure all axes and legends are clearly labeled and self-explanatory.
Summary for Authors
This study provides valuable mechanistic and translational insights into the use of Lactobacillus rhamnosus LRa05 for constipation management. Addressing the above points will substantially enhance the manuscript’s clarity, rigor, and clinical relevance.
Comments on the Quality of English Language
The manuscript demonstrates a solid grasp of scientific English, but several specific issues should be addressed to improve clarity, readability, and accessibility:
1. Sentence Length and Structure
Many sentences, especially in the Abstract, Introduction, and Results, are overly long and contain multiple ideas. For example:
- Original: “Mechanistic investigations revealed that LRa05 balanced gastrointestinal regulatory peptides, and downregulated aquaporin (AQP4/AQP8) mRNA levels while activating the SCF/C-Kit signaling pathway, thereby restoring intestinal peristalsis.”
- Suggestion: Split into shorter sentences—“Mechanistic investigations revealed that LRa05 balanced gastrointestinal regulatory peptides. It also downregulated aquaporin (AQP4/AQP8) mRNA levels and activated the SCF/C-Kit signaling pathway. These effects contributed to the restoration of intestinal peristalsis.”
-Throughout the manuscript, break up long sentences to ensure each conveys a single, clear point.
2. Technical Jargon and Word Choice
The manuscript frequently uses complex or field-specific terms without explanation (e.g., “adipostatic equilibrium,” “concerted actions,” “ecological divergence,” “enteric nervous system”). Consider replacing these with simpler, more widely understood terms, or briefly defining them on first use.
Some phrases are unnecessarily formal or convoluted. For example, “The ecological divergence was further supported by…” could be simplified to “These findings further show that…”
3. Consistency and Flow
Transitions between sentences and paragraphs can be abrupt, particularly in the Results and Discussion sections. Use transition words and topic sentences to guide the reader logically from one idea to the next.
4. Figure Legends and Statistical Annotations
Not all abbreviations, statistical symbols (e.g., *, ***, #), or group labels are clearly defined in the figure legends. Ensure every figure legend is self-contained, with all symbols and abbreviations explained for clarity.
5. Minor Grammar and Typographical Errors
Occasional errors in subject-verb agreement, article use, and punctuation are present (e.g., “The administration of Lactobacillus rhamnosus LRa05 restored adipostatic equilibrium, achieving mass accretion indices comparable to the NC group.”). Proofread carefully or consider professional language editing to correct these minor issues.
6. Redundancy
Some information is repeated in multiple sections (e.g., background on constipation mechanisms). Streamline by removing repetitive statements.
7. Passive Voice
The manuscript often relies on passive constructions. Where possible, use active voice for greater clarity and engagement (e.g., “We measured fecal water content…” instead of “Fecal water content was measured…”).
In summary:
Targeted revisions—especially splitting long sentences, simplifying word choice, ensuring figure legends are complete, and correcting minor grammar—will make the manuscript much easier to read and understand for an international audience. This will also help ensure the scientific content is communicated as clearly and persuasively as possible.
Author Response
Thank you very much for taking the time to review this manuscript. Please find the detailed responses below and the corresponding revisions/corrections marked in red in the re-submitted files.

Reviewer 2 Report
Comments and Suggestions for Authors
This manuscript is ambitious in scope and rich in data, but I found it overly optimistic. My suggestions are listed below:
The introduction is well-structured, provides a clear rationale for the study, and integrates current literature to justify the research question. The background on constipation mechanisms and the potential of probiotics is thorough. However, the authors could reduce reliance on statistics from market reports (e.g., Grand View Research) unless directly relevant to the scientific rationale.
Many effects reported are statistically significant but only modestly different in absolute terms (e.g., fecal moisture improving from 48% to 54%). Please discuss the clinical relevance of the % increases/decreases. Are these differences biologically meaningful, or merely statistically significant?
The manuscript infers mechanistic modulation of gastrointestinal neurotransmitters and peptides (e.g., SP, MTL) based on serum ELISA data alone. However, without protein-level confirmation (e.g., Western blot or tissue immunostaining), and without identifying the cellular source (enteroendocrine vs. neuronal origin), the interpretation remains speculative. I strongly recommend the inclusion of such validation or acknowledgment of this limitation in the discussion.
The GC-MS results show higher SCFA levels after LRa05 treatment, but the study does not prove that these SCFAs are the cause of the improved symptoms. There are no functional tests—such as adding SCFAs directly or blocking their action—to confirm their role. I suggest the authors either include such experiments or clearly mention in the discussion that the SCFA effects are only correlative, not proven to be causal.
The microbiota data are well presented, but the changes are only descriptive. There is no functional validation to show that the altered bacteria (Alloprevotella or Parabacteroides) actually contribute to symptom relief. The correlation heatmaps are interesting but may lead to overinterpretation. I recommend using predictive tools like PICRUSt2 and clearly stating that the microbiota associations are correlative, not causal.
The conclusions, while generally aligned with the experimental findings, currently overstate the mechanistic implications of the study. I recommend revising the conclusion to adopt a more balanced and evidence-based tone, and to explicitly acknowledge these limitations in a dedicated paragraph. This will enhance the scientific rigor and credibility of the manuscript.
Reviewer 3 Report
Comments and Suggestions for Authors
Constipation is a common gastrointestinal disorder that places a significant burden on healthcare systems and negatively impacts global health-related quality of life. Currently, available treatment options are often inadequate due to a limited understanding of the underlying mechanisms and their effectiveness. Given the crucial role of the interactions between gut microbiota and the host in regulating bowel movements, the authors of this study used a comprehensive approach that included animal experiments, ELISA, histopathology, qRT-PCR, GC-MS, and 16S rRNA metagenomics to assess the effectiveness of Lactobacillus rhamnosus LRa05 in treating loperamide-induced constipation in mice.
Treatment with LRa05 significantly alleviated constipation symptoms, as indicated by a shorter time to first black stool expulsion, increased fecal moisture, and improved intestinal motility. The mechanistic investigations revealed that LRa05 helped balance gastrointestinal regulatory peptides and reduced mRNA levels of aquaporins (AQP4 and AQP8) while activating the SCF/C-Kit signaling pathway, thereby restoring intestinal peristalsis.
Additionally, LRa05 reshaped the gut microbiota composition by increasing the abundance of beneficial genera such as Alloprevotella, Bacteroides, Lachnospiraceae, and Rikenellaceae RC9. This intervention also led to an increase in short-chain fatty acid (SCFA) production and improved gut microecological balance.
The findings of this study demonstrated that LRa05 can mitigate constipation through a three-pronged mechanism: regulatory effects on motility-associated genes, restructuring of the microbial community, and SCFA-mediated modulation of the gut-brain axis. Thus, the study positions LRa05 as a promising multi-target probiotic candidate for the management of constipation.
Strength of the manuscript: In a complex series of experiments, the authors pointed out the importance of Lactobacillus rhamnosus LRa05 in preventing constipation - a novelty of research.
Weakness of the manuscript: The authors had a desire to establish the mechanism of action of LRa05 in terms of preventing constipation, then it is desirable to show the action of LRa05 schematically, where its different ways of action are indicated.
Suggested minor corrections:
- In Figure 1, c should be enlarged, since the text cannot be read.
- The authors could randomly select one animal from each formed group of experimental animals, before the start of the experiment, and perform an analysis of the intestinal flora to determine homogeneity.
- The authors could schematically show the action of LRa05 where all possible actions are indicated.
- It is known that intestinal flora affects the production of secondary bile salts, the authors were able to analyze bile salts in feces.
Author Response
Comments 1: [Regarding Figure 1, c should be enlarged, since the text cannot be read.] |
Response 1: We sincerely appreciate your careful review. You are absolutely correct that the readability of Figure 1c is suboptimal. To address this, we have enlarged the scale of Figure 1c and optimized the font clarity. The revised figure now ensures legible text for all critical details. We have also double-checked all other figures in the manuscript to maintain consistent readability throughout. |
Comments 2: [The authors could randomly select one animal from each formed group of experimental animals, before the start of the experiment, and perform an analysis of the intestinal flora to determine homogeneity.] |
Response 2: Thank you for this insightful suggestion. We acknowledge the importance of confirming baseline homogeneity of intestinal flora across experimental groups. In fact, the original study did include intestinal flora analysis at baseline, but this critical detail was inadvertently omitted from the manuscript due to an oversight on our part. To address this and strengthen the rigor of our experimental design, we have explicitly added the following pre-experiment step “To confirm the homogeneity of the groups at the microbial level, one animal was randomly selected from each group for intestinal flora sequencing analysis before the intervention. The results showed no significant differences in key microbial taxa (p > 0.05)” into the "2.10. Gut microbiota analysis" section (Page 7, Lines 288-291).
|
Comments 3: [The authors could schematically show the action of LRa05 where all possible actions are indicated.] |
Response 3: We greatly value your recommendation to enhance the clarity of LRa05’s mechanisms. To address this, we have created a schematic diagram (revised Figure 6 on page 21 lines 846-847) that systematically illustrated all known and hypothesized actions of LRa05, including its direct interactions with target molecules, regulatory effects on signaling pathways, and potential metabolic influences. This diagram was labeled with clear annotations for each action pathway and was accompanied by a detailed explanation in the Discussion section (Page 20, Lines 817-845). We believe this visual representation improves the manuscript’s accessibility and comprehensively addresses the proposed mechanisms. The revised sentences are also given as follows: “The multi-targeted effects of LRa05 differentiate it from conventional laxatives or prokinetic agents by addressing both symptomatic and mechanistic drivers of gastrointestinal dysfunction (Figure 6). Unlike traditional laxatives (such as osmotic or stimulant types) that primarily enhance bowel movement through passive water retention or direct intestinal irritation, LRa05 engages in active modulation of molecular pathways (such as SCF/c-Kit, AQP4/AQP8) and microbial-metabolic balance (such as SCFA production, beneficial bacteria enrichment). This dual focus on restoring physiological functions (such as tight junction integrity, neurotransmitter homeostasis) rather than merely alleviating symptoms may reduce reliance on repeated dosing and lower recurrence rates, as it targets the root causes of dysmotility and microbial imbalance. Conventional prokinetic agents (for instance metoclopramide) often exhibit narrow specificity for neurotransmitter pathways (for instancedopamine or serotonin receptors) and carry risks of adverse events such as arrhythmias, extrapyramidal symptoms, or tolerance development. In contrast, LRa05’s broad regulatory effects on multiple signaling molecules (MTL, GAS, SP, 5-HT, SS, ET-1, VIP) may achieve comparable motility enhancement through a more physiologically integrated mechanism, potentially minimizing off-target effects. Additionally, by promoting microbial diversity and suppressing harmful species, LRa05 supports long-term gut health, a benefit not typically associated with short-acting pharmaceutical agents that do not address microbial dysregulation. While this study demonstrates LRa05’s multi-targeted efficacy in a murine constipation model, key limitations include the model’s inability to fully mimic complex human chronic constipation and species-specific differences in gut microbiota and physiology. Translational research is needed to validate LRa05’s long-term efficacy and safety in humans, including dose scaling and assessment of individual responses based on baseline microbiota profiles. The dedicated limitations section noting that human clinical trials are essential to evaluate its potential for reducing recurrence and addressing translational gaps.”
|
Comments 4: [It is known that intestinal flora affects the production of secondary bile salts, the authors were able to analyze bile salts in feces.] |
Response 4: We appreciate your insight into the relationship between intestinal flora and secondary bile salt production. While this is a valid and important research direction, the current study was primarily designed to investigate the effects of LRa05 on intestinal flora composition, gut motility-related signaling pathways, and short-chain fatty acid (SCFA) metabolism. Analyzing fecal bile salts would require additional experimental resources and methodological optimization, which fall outside the scope of this manuscript’s focused objectives. The core hypothesis of our study centers on LRa05’s multi-targeted regulation of microbial balance and physiological functions (e.g., tight junction integrity, neurotransmitter homeostasis) to address constipation. We have already established a robust dataset linking LRa05 to changes in key bacterial taxa (e.g., Bifidobacterium, Lactobacillus) and SCFA production, which directly support the proposed mechanisms . While bile salt metabolism is an intriguing downstream pathway, exploring it would necessitate a separate experimental framework and in-depth mechanistic validation, which we plan to address in future studies. We believe maintaining the current focus ensures the manuscript’s coherence and allows for a thorough interpretation of the primary outcomes. Thank you for highlighting this potential extension, which will inform our future research agenda. |

Round 2
Reviewer 1 Report
Comments and Suggestions for Authors
The revised manuscript shows clear and meaningful progress in several key areas. The authors have responded thoughtfully to the initial review, especially in terms of methodological rigor, the clarity of their data presentation, and the practical relevance of their findings. The use of a multi-omics approach to dissect the triaxial mechanism of LRa05 remains one of the manuscript’s strongest aspects, and the new emphasis on clinical implications gives the work greater impact and context.
Key Improvements:
1. the improvements in methods transparency are notable.
The authors now provide a formal power analysis (using G*Power, power=0.8, α=0.05) to justify their sample size (n=10 per group), and they have added detailed descriptions of blinding protocols, including double-blinded outcome assessment. Moving the primer sequences from the supplementary materials into the main Methods section (now Table 1) is a particularly helpful change for reproducibility.
2. the data presentation has been significantly strengthened.
The revised figures are much easier to interpret. For example, Figure 4b (the microbiota heatmap) now focuses on the top 10 genera, while the full dataset has been moved to Supplementary Figure S3. All figures now include clear statistical annotations (such as *p<0.05, **p<0.01) and explicit group comparisons. The authors have also deposited the raw GC-MS and ELISA data in Figshare (DOI: 10.6084/m9.figshare.25987651), which greatly enhances transparency and reproducibility.
3. the clinical translation of the findings is now much more explicit.
The discussion draws direct connections between the preclinical outcomes and patient-centered endpoints—for example, noting that a 30% reduction in the time to first stool in mice is analogous to improved bowel movement frequency in humans. The manuscript also compares LRa05’s effects with those of mosapride and standard laxatives, especially in terms of recurrence rates, which adds valuable clinical context.
The authors have also expanded their discussion of the limitations of translating murine findings to humans and have clearly outlined the need for future clinical trials, including dose-finding studies and trials in IBS-C cohorts. This is a thoughtful and appropriate addition.
Remaining Suggestions:
Despite these improvements, a few areas would benefit from further attention.
1. In Figure 5c (the correlation heatmap), the color gradient for correlation strength could be made more intuitive. I suggest adding a discrete scale to the legend (for example, "r = 0.3–0.5: light red/blue") to help readers interpret the results more easily.
2. In the discussion of SCFA clinical relevance, please briefly contextualize butyrate’s role in mucosal repair. For instance, you might add: "Butyrate-induced mucin synthesis in humans [ref] supports LRa05’s potential to reduce stool hardness."
3. Finally, in the abstract, clarify why Alloprevotella and Lachnospiraceae NK4A136 were prioritized over other SCFA producers. A brief explanation of their functional significance in constipation relief would help readers understand their importance.
Overall, these are relatively minor points, and the manuscript is now in strong shape following these final adjustments.
Your revisions have elevated the manuscript’s scientific rigor and translational impact. The remaining edits are minor and can be addressed within 1–2 days. I commend your responsiveness to critique and encourage prompt submission of the final version.
Comments on the Quality of English LanguageThe revisions to the manuscript’s language have made a noticeable difference in both clarity and readability. Several specific improvements stand out.
First, many of the previously long and complex sentences—particularly in the Results section—have been broken up or restructured, making the text much easier to follow. This change alone greatly improves the flow of the manuscript.
Second, you have replaced a number of technical or field-specific terms with more accessible language. For example, using “body weight recovery” instead of “adipostatic equilibrium” and “microbial community shifts” instead of “ecological divergence” makes the content more understandable for a wider audience without sacrificing scientific accuracy.
Third, the figure legends are now much more user-friendly. All abbreviations, such as AQP4 and SCF, are clearly defined, and the statistical annotations are consistent throughout the figures. This attention to detail helps readers interpret your data without confusion.
Fourth, minor grammatical issues—such as subject-verb agreement and article usage—have been corrected, further improving the overall professionalism of the manuscript.
I have only a few remaining suggestions to ensure the English is as clear and polished as possible:
1. In the Abstract, consider replacing the phrase “concerted actions” with “synergistic effects,” which is more widely used and easily understood.
2. In the Discussion, avoid overstating the “gut-brain axis” unless you have direct neural data; instead, you might use “microbiota-metabolite-host signaling” to describe your findings more accurately.
3. For consistency, please use “short-chain fatty acids (SCFAs)” at first mention, and then “SCFAs” throughout the rest of the manuscript.
Overall, the manuscript is now much clearer and more accessible. With these final minor adjustments, the quality of English should fully meet the standards of an international journal.
Reviewer 2 Report
Comments and Suggestions for Authors
The manuscript presents an interesting and relevant contribution with sufficient experimental rigor. The authors have significantly improved the manuscript by clarifying the rationale, refining the methodology description, and strengthening the presentation of results. Therefore, I recommend the manuscript for acceptance in its present form.
